# Precision ion separation via self-assembled channels

Shanshan Hong [1], Maria Di Vincenzo [2], Alberto Tiraferri[3], Erica Bertozzi [3], Radosław Górecki [2], Bambar Davaasuren[4], Xiang Li [1] & Suzana P. Nunes [1,2,5] ✉

Selective nanofiltration membranes with accurate molecular sieving offer a solution to recover rare metals and other valuable elements from brines. However, the development of membranes with precise sub-nanometer pores is challenging. Here, we report a scalable approach for membrane fabrication in which functionalized macrocycles are seamlessly oriented via supramolecular interactions during the interfacial polycondensation on a polyacrylonitrile support layer. The rational incorporation of macrocycles enables the formation of nanofilms with self-assembled channels holding precise molecular sieving capabilities and a threshold of 6.6 ångström, which corresponds to the macrocycle cavity size. The resulting membranes provide a 100-fold increase in selectivity for $Li^+/Mg^{2+}$ separation, outperforming commercially available and state-of-the-art nanocomposite membranes for lithium recovery. Their performance is further assessed in high-recovery tests under realistic nanofiltration conditions using simulated brines or concentrated seawater with various $Li^+$ levels and demonstrates their remarkable potential in ion separation and $Li^+$ recovery applications.

The exponential growth of the semiconductor industry and the urgent demand for global decarbonization hinge on securing a steady and sustainable supply of valuable metals necessary for manufacturing modern energy storage systems. Furthermore, recent events like the pandemic and geopolitical crises have shown how easily the supply chains of components and essential elements for electronics can be disrupted. The access to element resources has become a question of security for many countries. Effective and sustainable methods for the recovery of metal ions from different sources including e-waste recycling are increasingly important. Among the elements with the most prominent visibility is lithium. The rapid dissemination of electric vehicles relies on high-energy-density lithium-ion batteries[1].

Resources are limited. While 25% of lithium is found in hard rock ores, natural brines, such as salt lakes, seawater, and geothermal resources account for approximately 60% of retrievable lithium on Earth[2]. Lithium extraction from ores requires particle squashing and acid dissolving. In contrast, direct extraction from brine solutions is more efficient and economical. But in this case, the main challenge is the fact that lithium is not available as a single element, but as a minor component with a much higher concentration of other metal ions such as magnesium and sodium. To facilitate the separation and overcome chemically intensive precipitation processes[3], research on alternative sustainable methods is being conducted worldwide[4].

Membrane technology is under consideration as a prominent green approach for ion separations and recovery, including the

[1]Chemistry Program, Physical Science and Engineering Division (PSE), King Abdullah University of Science and Technology (KAUST), Thuwal, Saudi Arabia. [2]Environmental Science and Engineering Program, Biological and Environmental Science and Engineering Division (BESE), King Abdullah University of Science and Technology (KAUST), Thuwal, Saudi Arabia. [3]Department of Environment, Land and Infrastructure Engineering (DIATI), Politecnico di Torino, Corso Duca degli Abruzzi 24, Turin 10129, Italy. [4]Core Labs, King Abdullah University of Science and Technology (KAUST), Thuwal, Saudi Arabia. [5]Chemical Engineering Program, Physical Science and Engineering Division (PSE), King Abdullah University of Science and Technology (KAUST), Thuwal, Saudi Arabia. ✉e-mail: suzana.nunes@kaust.edu.sa

preconcentration and lithium extraction from lithium-bearing brines. Nanofiltration (NF) membranes could play a major role. However, the state-of-the-art membranes synthesized by interfacial polymerization do not enable precise molecular sieving[3]. Different approaches have been investigated to improve ion selectivity. A primary one to enhance the separation of $Li^+/Mg^{2+}$ is by tuning the surface charge density of the membrane active layer to enhance the electrostatic-based removal of $Mg^{2+}$[,5]. In charged systems, the ions' mass transfer is controlled by a combination of Donnan exclusion, steric hindrance, and dielectric exclusion[5]. To improve membrane selectivity, considerable research efforts have been placed towards exploring novel materials with ordered, uniform pore structures, as well as tunable functionalities. Examples of such materials include covalent-organic framework nanosheets[6], metal-organic framework nanocrystals[6,7], two-dimensional nanosheets[8,9], and zero-dimensional nanoparticles[10]. However, the incorporation of these structures into scalable membrane materials has been hindered by a series of drawbacks that vary from inherent poor stability[11] and lack of chemical compatibility with the surrounding polymeric matrix of the selective layer to the availability of the selective component, and the feasibility of membrane fabrication on a large scale. Therefore, the practicality of many approaches for the recovery of metals from complex water feed streams remains limited.

Recent advances in nanotechnology and supramolecular chemistry have opened new prospects for developing nanostructured membranes with preformed pores using straightforward synthetic water and ion channels[12–14]. The self-assembly of building blocks into channels can be achieved through noncovalent bonds, such as hydrogen bonding, electrostatic interactions, host-guest interactions, or halogen bonding, to form precise pores with fine functionalities[13,15,16]. We have proposed and demonstrated that the integration of supramolecular assemblies, macrocycles, and porous organic cages into membrane selective layers can be effectively achieved if amino functionalities are available for cross-linking by interfacial polymerization[10,17–19]. This method leads to scalable and robust thin-film composite membranes and the procedure can be reproduced in typical roll-to-roll manufacturing machines. The previous membranes we manufactured in this way were tested for organic solvent nanofiltration. The membranes feature crosslinked unities with hydrophobic inner cavities and hydrophilic outer surfaces. We propose that an analogous strategy could be efficient to produce membranes with a high density of ion channels.

Herein, we report the successful development of ultrathin cyclodextrin-based polyamide membranes with self-assembled ion channels for accurate molecular separation. This study provides a comprehensive understanding of the kinetic formation of layers, wherein precise cavity packing is achieved through supramolecular interactions. Empirical findings demonstrate the exceptional performance of our membranes, outperforming both traditional membranes and previously reported nanocomposite membranes. Remarkably, these hybrid nanofilms enable the transport of $Li^+$ through the channels, while rejecting divalent cations such as $Mg^{2+}$. Their performance in $Li^+/Mg^{2+}$ separation is assessed under realistic nanofiltration conditions using simulated brines. This strategy is particularly significant as it allows the large-scale production of defect-free membranes for precise ion separation via a straightforward interfacial polymerization reaction.

## Results

### Cyclodextrin-based membranes incorporating self-assembled ionic channels

For the fabrication of thin composite film membranes, an asymmetric porous support is usually soaked with an aqueous solution containing amino-functionalized monomers and then brought in contact with an immiscible organic phase containing acid chlorides. A reaction between monomers happens only at the interface forming the membrane polyamide selective layer, as depicted in Fig. 1a. In this work, cyclodextrins with 6, 7, or 8 glucose subunits were amino-functionalized (Supplementary Figs. 1–3) and dissolved in the aqueous phase (fabrication conditions in Supplementary Table 3). Heptakis(6-deoxy-6-amino)-β-cyclodextrins (β-cyclodextrin, Am7CD) have limited solubility in water unless the amino groups are partially protonated, e.g., under acidic conditions, which would eventually disrupt the supramolecular hydrogen bonding network[20]. Once protonated, the water-soluble Am7CD heptahydrochloride salt (Am7CD•7HCl) was dissolved in lithium hydroxide, resulting in a clear solution (LiOH-Am7CD solution, Supplementary Fig. 4). We observed that in this way, the Am7CD molecules in aqueous solution self-assembled into an ordered arrangement, exhibiting visible lattice fringes confirmed by cryogenic-transmission electron microscopy (Cryo-TEM) (Fig. 1b). The packing pattern of cyclodextrin in water typically varies depending on factors such as cavity size, type of guest molecules, and concentration[21–23]. In the present study, single-crystal X-ray diffraction was used to examine the crystal structure of the obtained crystals by allowing the sealed solution to rest for 48 h (Supplementary Fig. 5). The Am7CD molecules are functionalized with seven amino groups on the primary side (narrow rim), and fourteen hydroxyl groups on the secondary side (wider rim). Strong intermolecular hydrogen bonding is established between hydroxyl and amino groups, with D-A distance of ~2.7 Å (Supplementary Table 1), resulting in 1D channel-like arrangement of Am7CD molecules packed along [100] (Fig. 1c and Supplementary Fig. 6).

Macrocycle ion channels have been previously reported embedded in lipid bilayers[24–26]. Here, however, the amino groups of Am7CD assemblies react with trimesoyl chloride (TMC) via the classical interfacial polymerization procedure. The highly cross-linked polyamide membranes incorporate a high density of self-assembled ionic channels (Fig. 1a). The formation of the polyamide active layer was confirmed by Fourier transform infrared (FTIR) spectroscopy (Supplementary Fig. 7) and X-ray photoelectron spectroscopy (XPS) (Supplementary Fig. 8). Scanning electron microscopy (SEM) (Supplementary Figs. 9–11) and atomic force microscopy (AFM) (Supplementary Figs. 12 and 13) revealed a smooth and continuous defect-free active layer on the ultrafiltration polyacrylonitrile (PAN) support layer.

Freestanding selective layers were further characterized by grazing-incidence wide-angle X-ray scattering (GIWAXS) (Fig. 1d) to confirm the incorporation and order of the channels embedded in the membrane structure. Notably, the GIWAXS analysis indicates the presence of polycrystalline species and small crystallites over the membrane selective layer. During the interfacial polymerization process, self-assembled Am7CD unities were cross-linked forming an ultrathin selective layer. AFM analysis corroborates these results and shows that the crystals were uniformly distributed throughout the membrane (Fig. 1e), while TEM images of the polyamide selective layer reveal long-range ordering of these crystals with a distance of 2.53 Å between the adjacent lattice fringes (inset of Fig. 1f, scale bar 1.5 nm). The diffraction peak in Fig. 1g appearing at 34.8° is relative to the (310) reflection of the Am7CD crystal and corresponds to periodic distance similar to that imaged by TEM for the assemblies in membranes. Furthermore, the good agreement between the GIWAXS in-plane and out-of-plane scattering profiles and the powder X-ray diffraction of the self-assembled Am7CD polycrystalline powder, in conjunction with the calculated XRD pattern, indicates the successful integration of the subnanometer channel structure within the membrane (Supplementary Fig. 14). The crystal theoretical data is presented in Supplementary Table 2.

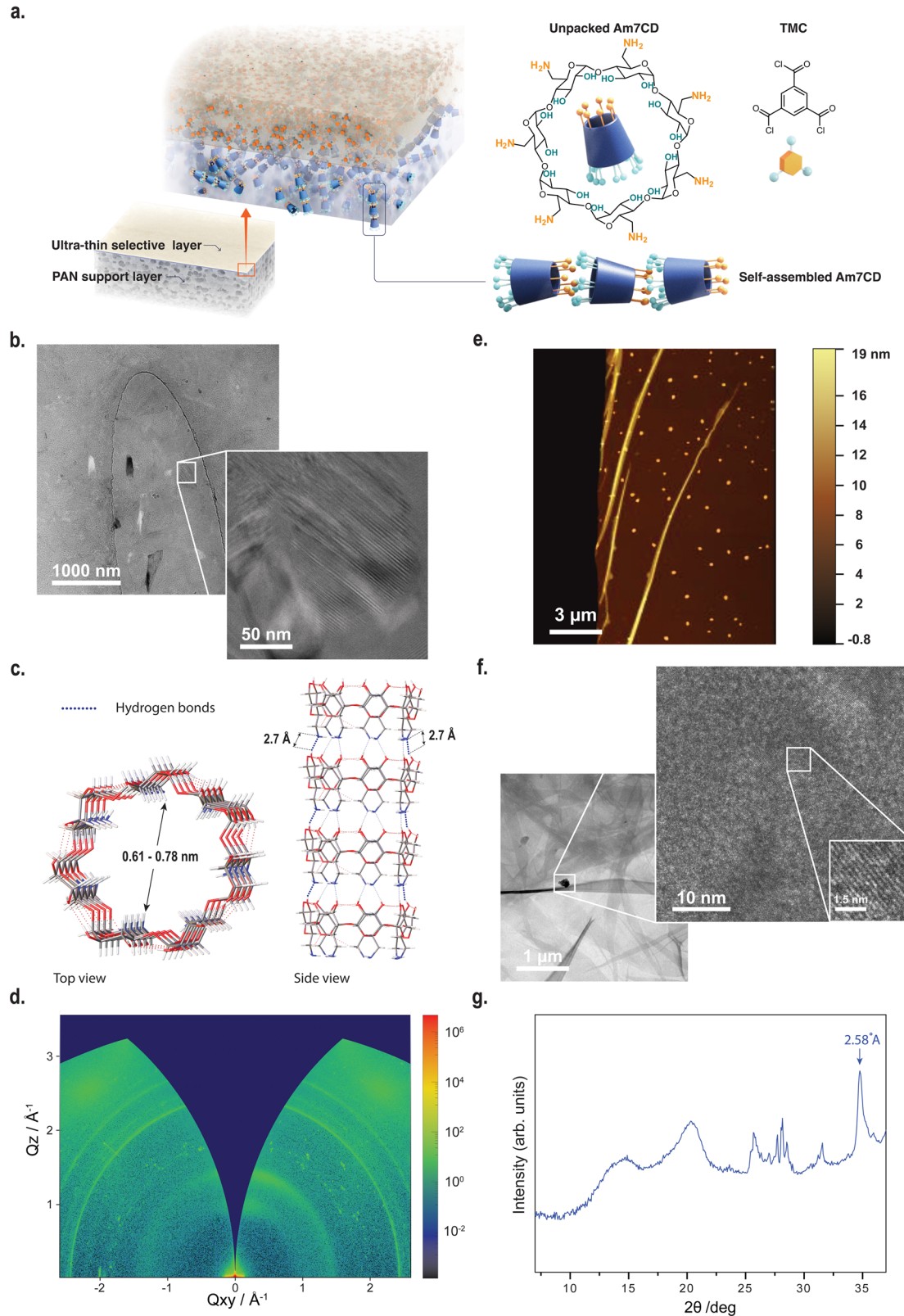

**Fig. 1 | Fabrication and characterization of Am7CD membranes incorporating artificial ion channels. a** Schematic illustration of LiOH·Am7CD·TMC membranes fabrication via interfacial polymerization. **b** Cryo-TEM micrographs of a 1.23%wt Am7CD•7HCl/0.03 M LiOH solution. Inset: high magnification image of assemblies. **c** Molecular packing of the Am7CD single crystal[45] (top and side view). **d** Two-dimensional X-ray diffraction pattern of Am7CD membranes (LiOH-Am7CD-0.01 TMC). **e** AFM topographic image of the Am7CD membrane (LiOH-Am7CD-0.01 TMC) with dispersed crystals transferred onto a silicon wafer. **f** TEM images of Am7CD membranes with incorporated assemblies (LiOH-Am7CD-0.05 TMC), highlighting a high magnification detail. Inset (with a scale-bar of 1.5 nm): the magnification of a selected area showing distinct lattice fringes. **g** GIWAXS integrated one-dimensional pattern of Am7CD membranes (LiOH-Am7CD-0.01 TMC).

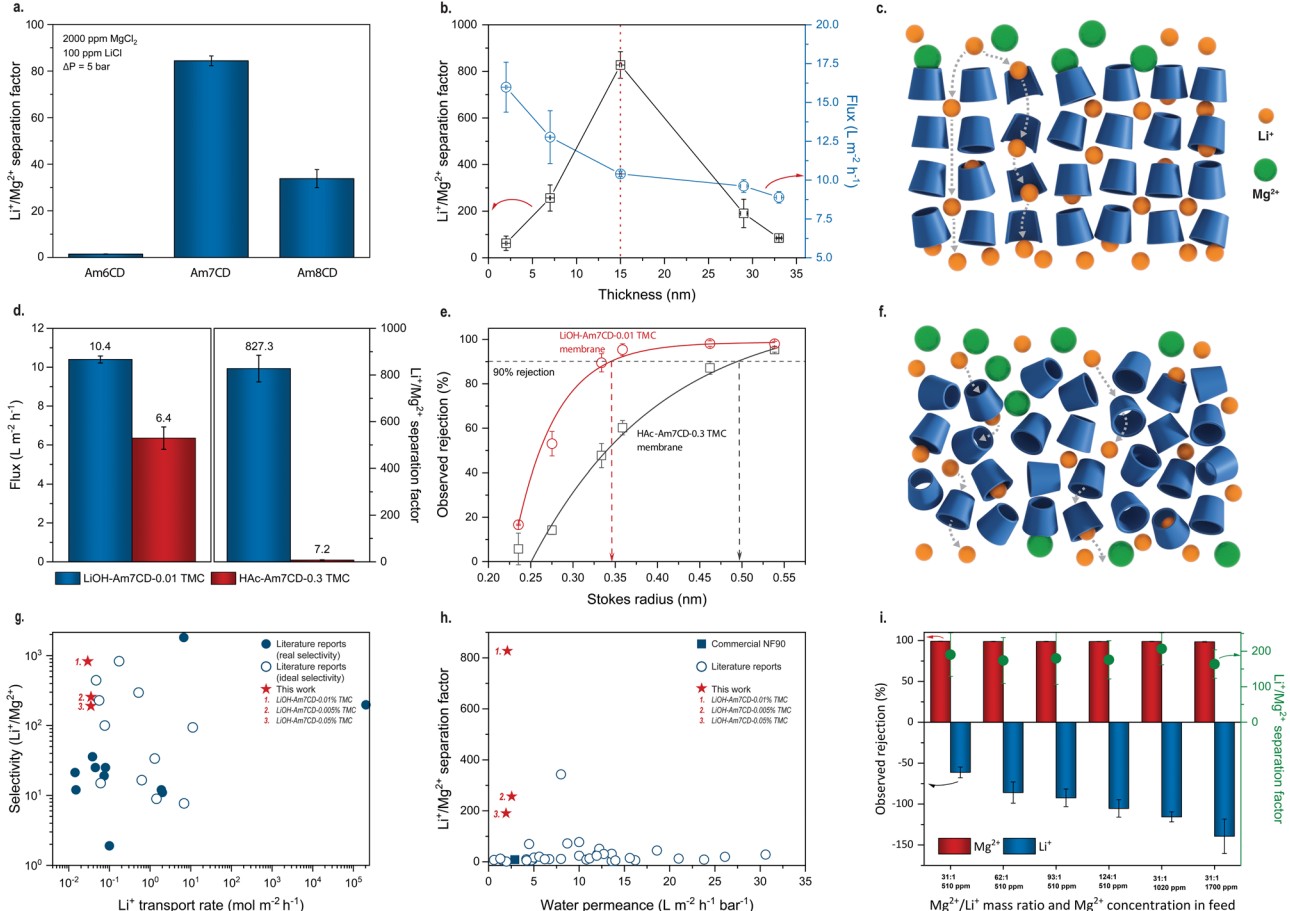

**Fig. 2 | Molecular sieving performance of cyclodextrin membranes for Li⁺/Mg²⁺ separation. a** Selectivity comparison between amino-cyclodextrin membranes with 6, 7, and 8 (LiOH-AmCD-0.1 TMC) glucose subunits; Feed solutions contain 2000 ppm MgCl₂ and 100 ppm LiCl. **b** Selectivity and flux of LiOH-Am7CD-TMC membranes with various thicknesses; Feed solutions contain 2000 ppm MgCl₂ and 100 ppm LiCl. **c** Schematic diagram illustrating the ion transport through the nanofilms with the CD-stacked channel structure. **d** Comparison of separation factor (right) and flux (left) between LiOH·Am7CD-0.01 TMC and HAc-Am7CD-0.3 TMC membranes with binary salt feed compositions. Feed solutions contain 2000 ppm MgCl₂ and 100 ppm LiCl. **e** Rejection of neutral solutes with different Stokes radius by LiOH-Am7CD-0.01 TMC and HAc-Am7CD-0.3 TMC membranes. The dashed lines indicate the sugar molecules that were rejected with a rate of 90% or higher. **f** Schematic diagram illustrating the tortuous transport through the nanofilms without the CD-stacked channel structure. **g** Comparison of Li⁺/Mg²⁺ selectivity and Li⁺ permeation rate between LiOH-Am7CD-0.05/0.01/0.005 TMC membranes and other reported membranes containing COFs, MOFs, and other 2D materials. **h** Permeance and Li⁺/Mg²⁺ separation factor of LiOH-Am7CD-0.05/0.01/0.005 TMC membranes, commercial NF 90 membranes, and other reported pressure-driven nanofiltration membranes. (Note: selected membranes are given in Supplementary Tables 6 and 7). **i** Ionic rejection and corresponding Li⁺/Mg²⁺ separation factor of LiOH-Am7CD-0.05 TMC membranes using a series of feeds with different Mg²⁺/Li⁺ mass ratios and Mg²⁺ mass concentration. (Note: all tests were conducted at 5 bar, except for the final one with a Mg²⁺ mass concentration of 1700 ppm, which was conducted at 10 bar to overcome the osmotic pressure). Error bar standard deviation.

## Precise Li⁺/Mg²⁺ separation

The selectivity of the active layer is essential to achieve an accurate ion separation[5,27,28]. By polymerizing functionalized cyclodextrins with different sizes of the intrinsic cavity, we can precisely tune the diameter of the transport channel (Supplementary Figs. 15 and 16). The performance of Am7CD-based membranes was compared with that of membranes prepared by incorporating Am6CD and Am8CD structures, which have 6 and 8 glucose subunits, respectively. Results show that Am7CD membranes exhibited the highest selectivity among the three (Fig. 2a and Supplementary Figs. 17–19). The Am6CD membranes exhibited the lowest separation factor and rejection of MgCl₂, probably due to the formation of a defective layer caused by the insufficient reactivity in competing with water to react with TMC (Supplementary Fig. 10). Larger cyclodextrin cavities (Am8CD) enhanced flux but compromised the selectivity. Therefore, Am7CD-based membranes had the optimum performance which may be attributed to their optimal cavity size towards the target solutes and the sufficient reactivity with TMC during synthesis.

A recent work from the Livingston group demonstrates that reducing the membrane thickness facilitates orientated packing and precise molecular sieving[29]. In the case of Am7CD membranes, in our work, the thickness was controlled by varying the TMC concentration during fabrication, ranging from 2 to 33 nm (Supplementary Fig. 12). In Fig. 2b, as the thickness increased to 15 nm, the selectivity followed a direct relationship, while the flux exhibited an inverse trend, which is a typical trade-off observed in nanofiltration membranes. However, as the thickness was further increased, the selectivity decreased, following a pattern similar to other reported studies[29]. Moreover, higher TMC concentrations led to the formation of defects, resulting in lower overall MgCl₂ and LiCl selectivity (Supplementary Figs. 20–22).

The membrane characterization results imply the successful incorporation of ordered self-assembled channels when adding LiOH to the Am7CD solution. We surmise that the seamless incorporation of

assembled ionic channels is essential to increase the nanofiltration performance of cyclodextrin membranes for $Li^+/Mg^{2+}$ separation. Well-confined sub-nanometer channels should provide precise molecular sieving for solutes with similar sizes, while the direct pore structure reduces the transport distance and enhances the permeation (Fig. 2c). To verify these hypotheses, membranes without channels were fabricated using an aqueous solution in which acetic acid was added to dissolve synthesized Am7CD molecules at lower pH. For these membranes (referred to as HAc-Am7CD-TMC membranes), TEM images (Supplementary Fig. 23) revealed the absence of ordered assemblies seen as crystalline particles across the selective layer. Furthermore, GIWAXS scattering profiles of the resulting membranes showed no obvious peaks (Supplementary Fig. 24), implying the absence of an ordered structure.

For a comparative study, we conducted nanofiltration tests under the same experimental conditions. In binary salt nanofiltration tests, the membranes with crystalline particles provided an impressive selectivity that was over 100-fold higher than that observed with the non-ordered membranes (827 compared to 7.2), along with an approximate 60% enhancement in flux for the membranes prepared with LiOH (Fig. 2d). The results of rejection experiments with neutral solutes (Fig. 2e) suggest that membranes prepared with LiOH had a cut-off rejection of roughly 0.34 nm, while those prepared with HAc exhibited a cut-off rejection of approximately 0.49 nm, which deviates from the cavity diameter of Am7CD (0.6 nm)[30]. Note that membranes prepared with HAc and LiOH had similar crosslinking degree, surface morphology, thickness, and surface charge (Supplementary Figs. 8–13, 25 and 26). Therefore, the simultaneous improvement in both permeance and selectivity may be attributed to the ordered channel nanostructure. The presence of randomly packed macrocycle molecules may instead generate irregular and less-selective intermolecular voids, enabling $Mg^{2+}$ ions to pass through the membrane. The resulting tortuosity would hinder the transport rate of molecules, resulting in lower flux (Fig. 2f).

Benefiting from well-structured channels, LiOH-Am7CD-TMC membranes showed excellent selectivity, comparable to those of novel materials, such as COFs, MOFs, and other 2D materials with uniform pore structure or interlayer spacing. The data from previous reports, plotted in Fig. 2g and related to various membranes containing 2D materials, were obtained in diffusion experiments using a standard setup where the rate of $Li^+$ transport was influenced by the concentration of the ionic solution or the applied voltage. However, in this study, the membranes were evaluated in a pressure-driven process. Therefore, for comparison with other nanofiltration membranes, we collected the available literature data for nanofiltration membranes applied to $Li^+/Mg^{2+}$ separation under analogous operating conditions. Additionally, a commercial NF 90 membrane (DuPont - Film Tech) was evaluated in our laboratory-scale system and the results are depicted as a square blue symbol in Fig. 2h. Remarkably, LiOH-Am7CD-0.01 TMC membranes proposed in this study outperformed most of the previously reported membranes, exhibiting superior selectivity (>800) and moderate permeance. We believe that permeance could be still enhanced by optimizing the choice of porous substrates in the future. Furthermore, LiOH-Am7CD-0.05 TMC membranes had a high performance even when operating with feeds of increasing $Mg^{2+}/Li^+$ ratios and $Mg^{2+}$ concentrations, providing a remarkably stable selectivity and a feasible productivity (Fig. 2i and Supplementary Fig. 27).

## Ion transport mechanism in Am7CD membranes

To gain further insights into the mechanism responsible for the high $Li^+/Mg^{2+}$ selectivity provided by the membranes in this study, we conducted a series of nanofiltration tests with LiOH-Am7CD-0.05 TMC membranes using feed solutions containing chloride salts. As the ionic solutions shared the same anion ($Cl^-$) (Supplementary Fig. 28), the

observed difference in rejection rates may be attributed primarily to the properties of the cations. The measured surface zeta potential indicates a negative charge of the membrane surface when the pH exceeds 4 (Supplementary Fig. 25). Therefore, the rejection rates followed the typical sequence of negatively charged nanofiltration membranes[31], which is $Na_2SO_4 > MgSO_4 > MgCl_2 > NaCl$ (Supplementary Fig. 29). We speculate that the high $Li^+/Mg^{2+}$ selectivity of membranes in this study was not governed by Donnan exclusion phenomena. The retention of uncharged model solutes revealed a threshold at 0.66 nm (Supplementary Fig. 16). Therefore, bulky ions such as $(Me)_4N^+$ and $(Et)_4N^+$, which are characterized by a rigid molecular size exceeding this limit, were effectively rejected (as shown in Fig. 3b). In contrast, most $H^+$ ions with a small, hydrated diameter of approximately 5.6 Å passed through the membrane (Fig. 3c). Notably, for ions with an ionic diameter smaller than the cyclodextrin cavity but characterized by a larger hydrated diameter, the observed rejection of the metal cations correlated well with their hydration energy (Fig. 3d): the energy barrier of ion dehydration relates to the free energy of hydration[32].

As a result, ions with lower hydration energy ('looser' hydration layer) suffer a lower energy penalty to deform or to partially strip the water shell layer, which is a necessary mechanism to pass through the channels. In contrast, ions with a higher hydration energy having a more rigid water shell must overcome a significant energy barrier to enter the channel[33] (Fig. 3a). In our study, the experimental linear correlation between observed rejection and hydration energy indicates that steric dehydration occurred at the channel entrance, further suggesting that the ion transport in Am7CD membranes may be predominantly governed by the hydration energy during ion partitioning. Therefore, $Mg^{2+}$ ions with a 'tighter' hydration shell undergo more hindrance than $Li^+$ ions to enter the channel and diffuse across the membrane. Furthermore, the easier dehydration of $Li^+$ would bolster the electrostatic attraction to the membrane surface[34], facilitating $Li^+$ transport and further enhancing selectivity.

## $Li^+/Mg^{2+}$ separation from realistic brines

The incorporation of self-assembled ion channels can have a fundamental role in enhancing the nanofiltration performance (Fig. 4a). Am7CD membranes were evaluated under cross-flow filtration conditions at varying pressures (5, 10, 18 bar) using simulated salt lake brine as feed solution[35]. Our membranes exhibited an average water permeance of $3.7 \, L \, m^{-2} \, h^{-1} \, bar^{-1}$ and provided exceptional $Li^+/Mg^{2+}$ separation factors, which increased with increasing pressure. Each collected feed and permeate sample was analyzed using inductive plasma coupled with optical emission spectroscopy (ICP-OES) to determine the single ion observed rejections. Specifically, Am7CD layers provided rejection rates for $Mg^{2+}$ equal to 97.7% and 99.5% at 5 and 18 bar, respectively, and maintained similar performance for $Ca^{2+}$. The composition of the feed solution is in Supplementary Table 4. Notably, the selective layer facilitated the transport of $Li^+$, resulting in outstanding separation factors of 129.8 and 184.1 for $Li^+/Mg^{2+}$ and $Li^+/Ca^{2+}$, respectively. These findings corroborate the remarkable potential of our membranes in ion separation and $Li^+$ recovery applications.

The performance of Am7CD membranes was also assessed for $Li^+/Mg^{2+}$ separation using three synthetic feed solutions that closely mimic the composition of real lithium-bearing brines characterized by various $Li^+$ levels. The composition of the synthetic brines are in Supplementary Table 5. Specifically, we used concentrated seawater (typical brine from RO seawater desalination processes)[36], Lungmu Co salt-lake brine[37], and Imperial geothermal brine[38]. These sources are interesting to study because their lithium concentration is sufficiently high to justify a pressure-driven process for its recovery and for the potential recovery of other metals. Also, the sources are different in terms of composition and cation-to-lithium ratios, thus allowing assessment of

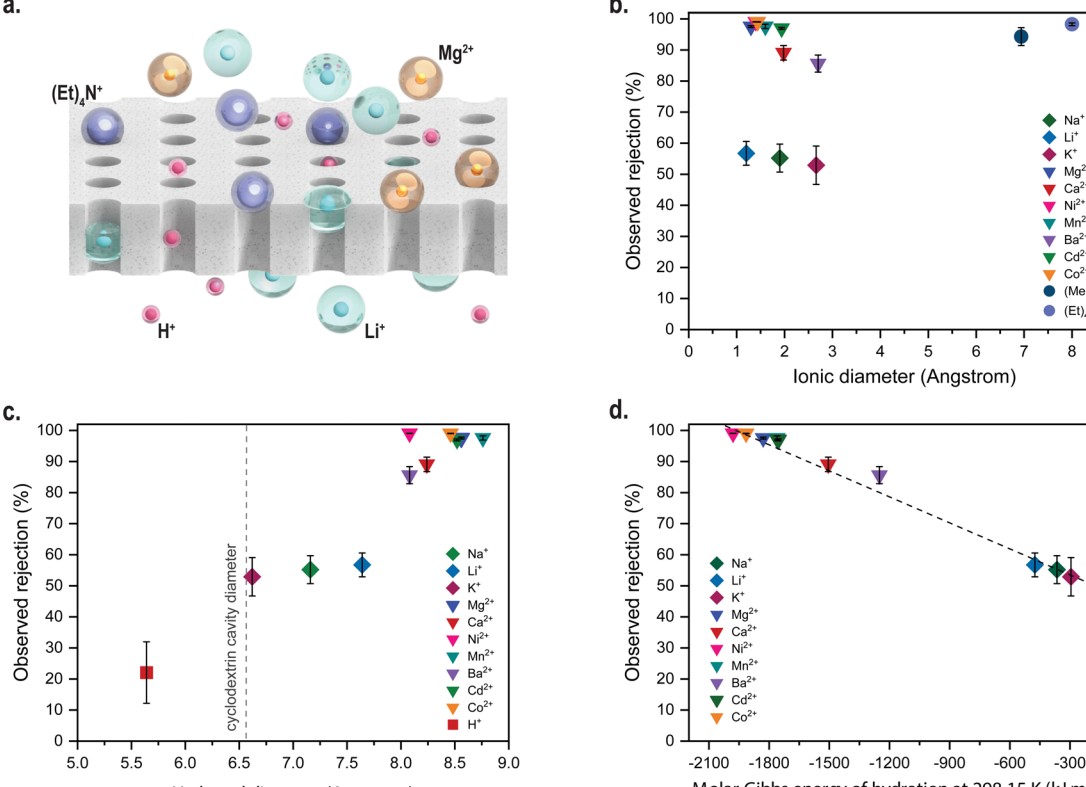

**Fig. 3 | Separation mechanism of the selective active layer and sieving properties of the membranes. a** Schematic diagram illustrating different ions passing through or rejected by ordered LiOH-Am7CD-0.05 TMC membranes. **b–d** Observed rejection versus (**b**) ionic diameter[46], (**c**) hydrated diameter[46], and **d** molar Gibbs energy of hydration[47] of different cations (with Cl⁻ as the counter-ion in the solution). Filtration tests were performed using a series of single-salt feed solutions with a concentration of 2000 ppm. The dash line is simply a guide for the reader. Error bar standard deviation.

the membrane behavior when challenged by diverse feed streams. Separation experiments were performed adjusting the operating conditions individually for each synthetic feed solution. The cross-flow rate for all tests was set at 5 L/h, while the applied pressure varied according to the specific synthetic solution, namely, 70 bar for concentrated seawater, and 60 bar for Lungmu Co salt-lake brine and Imperial geothermal brine (the detailed separation protocol is described in SI). Specifically, the membranes were evaluated in high-recovery filtration tests, thus simulating the single ions permeation and flux profiles that would be achieved in a single-stage nanofiltration plant under conditions representative of full-scale operation. For the concentrated seawater, the filtration was interrupted once over-saturation was reached, and salt precipitation started to occur (approximately from a recovery rate higher than 20-25%). Instead, for the other two brines, the endpoint was set when one of two conditions was reached first, either a recovery rate of 50% or fluxes lower than $3\,L\,m^{-2}\,h^{-1}$ (Fig. 4c–h).

The data in Fig. 4c–e show the trends of single ion observed rejections as a function of recovery: for concentrated seawater and Lungmu Co brine, the rejection of $Mg^{2+}$ remained consistently high and close to 95% throughout the entire filtration process, while a progressive increase of $Li^+$ transport was observed for recovery rate values between 10% and 20-25% (Fig. 4c, d). The rejection sequence was $Mg^{2+} > Ca^{2+} > Li^+ > Na^+ > K^+$, analogously to what has been reported to other membranes tested for lithium recovery in the literature[39–41].

As depicted in Fig. 4i, j, the separation factors for concentrated seawater and salt lake brine were similar for $Li^+/Mg^{2+}$ and $Li^+/Ca^{2+}$, whereas a different trend was observed with the geothermal brine. In the latter case, the high concentrations of $Ca^{2+}$ ions make scaling more

likely to occur during operation, impacting the overall rejection performance of the membrane. The lower observed rejections and separation factors with the geothermal brine, especially for divalent ions, may also be attributed to a possible screening effect along with a modification of the selective layer, resulting in an alteration of the effective pore size and in a reduction of the size sieving contribution[42]. Furthermore, the feed stream composition is complex and could lead to a multitude of interplaying factors driving ion separation performances[28,43], including various specific ion properties or thermodynamic barriers to transport, such as entropy-enthalpy compensation induced by ion-membrane interactions[44].

In all tests, as the feed solution became increasingly concentrated, the flux profile exhibited a gradual decline from its initial value due to the reduction in the driving force for filtration (Fig. 4f–h). Overall, results obtained in filtration tests simulating full-scale operation confirmed the promising separation performance for $Li^+/Mg^{2+}$ provided by the membranes proposed in this study using various brines as feed, which could be applied for metal recovery from concentrated seawater or salt-lake brine. Furthermore, it is reasonable to expect that enhanced fluid dynamics in real membrane elements would mitigate the impact of concentration polarization compared to laboratory-scale filtration, resulting in higher fluxes and potentially higher separation factors.

## Discussion

The interfacial polymerization of amino cyclodextrin and TMC resulted in the fabrication of ultrathin and robust composite membranes with precise porosity. Characterization confirmed the formation of selective layers incorporating nanochannels originating from the supramolecular assembly of Am7CD macrocycles.

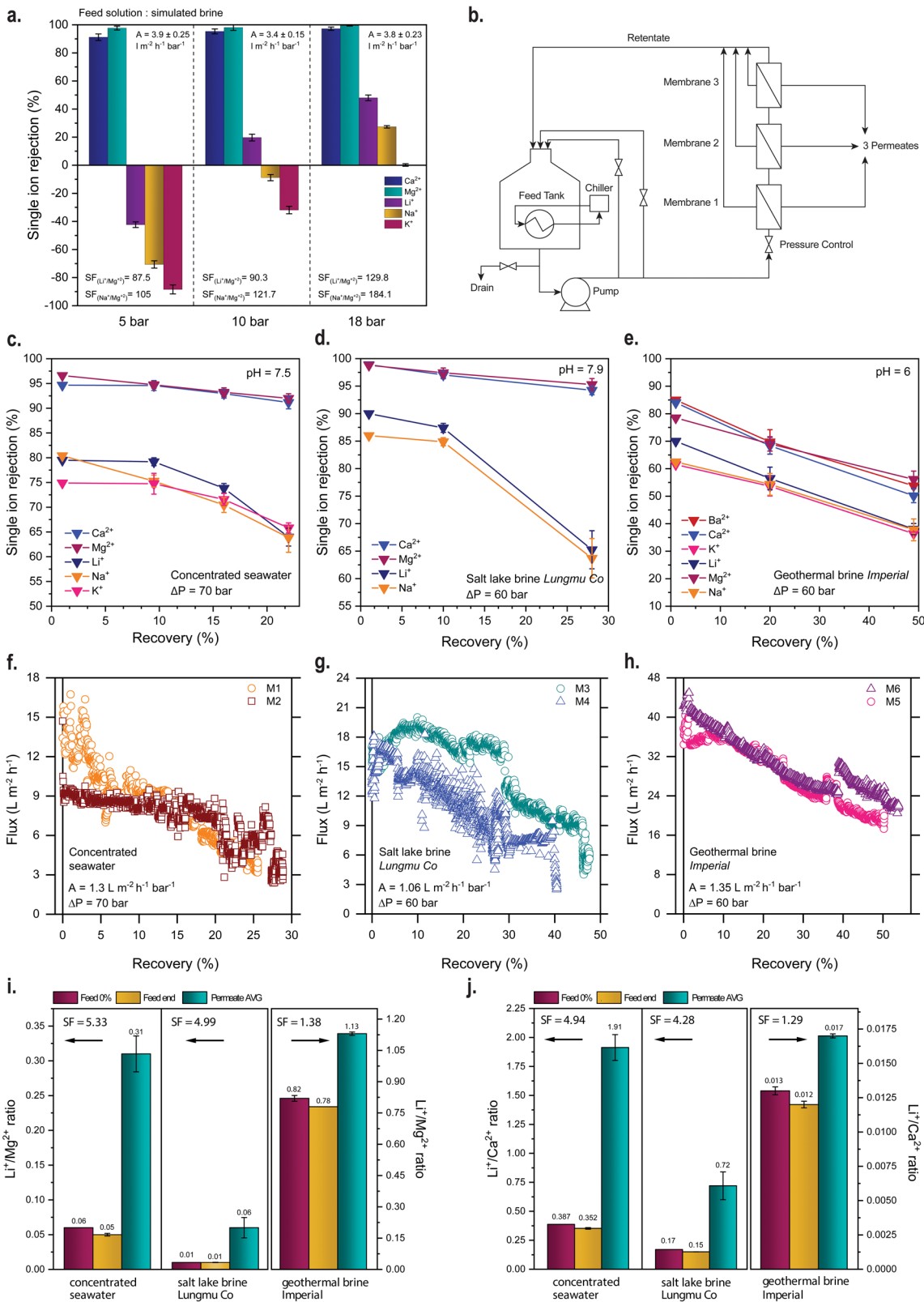

Experimental findings unveiled that self-assembled Am7CD formed ionic channels that had the macrocycle cavity size, enabling exceptional $Li^+/Mg^{2+}$ separation through a dehydration mechanism. Filtration tests further revealed that our hybrid active layers exhibit superior selectivity, outperforming commercially available and literature-reported nanocomposite membranes. Ultimately, we presented the practical viability of effectively sieving $Li^+$ ions from various synthetic feed solutions that closely mimic the composition of real lithium-bearing brines under conditions representative of full-scale operation. This study provides a promising approach to fine-tuning the membrane structure and porosity for precise molecular separations.

**Fig. 4 | Ion separation performance of cyclodextrin membranes under cross-flow filtration conditions using synthetic real lithium-bearing brines.**
**a** Experimental single ion observed rejection, water permeance (A), and separation factor (SF) of LiOH·Am7CD-0.05 TMC membranes. Filtration conditions: 5 bar, 10 bar, and 18 bar applied pressure using simulated salt-lake brine as feed solution[35] (the ionic composition of the feed solution is reported in Supplementary Table 4); **b** Schematic flow diagram of the cross-flow laboratory scale filtration system. **c–e** Experimental single ion observed rejection and **f–h** water flux of M1-M6 membranes (all LiOH·Am7CD-0.05 TMC membranes) plotted as a function of the recovery using three different synthetic brine solutions with various Li+ levels,

namely, concentrated seawater[36], Lungmu Co salt-lake brine[37], and Imperial geothermal brine[38]; here, the filtration conditions were: 70 bar for concentrated seawater (pH 7.5), and 60 bar for Lungmu Co salt-lake brine (pH 7.9) and Imperial geothermal brine (pH 6) (the ionic compositions of the feed solutions are listed in Supplementary Table 5); A is the intrinsic water permeability. The observed rejection and the retentate stream ionic concentrations as a function of the recovery are presented in Supplementary Figs. 30–32; Experimental **i**, Li+/Mg2+ and **j**, Li+/Ca2+ mass ratios in the synthetic feed solutions (at 0% recovery and at the end of recovery, respectively) and in the final collected permeate. Error bar standard deviation.

## Data availability
Data generated and analyzed in this study are included with this Article and the Supplementary Information. Additionally, they are accessible from the corresponding author upon request. Crystallographic data for the structure reported in this Article have been deposited at the Cambridge Crystallographic Data Centre, under deposition numbers CCDC 2290245. Copies of the data can be obtained free of charge via https://www.ccdc.cam.ac.uk/structures/.

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

## Acknowledgements

This work was sponsored by King Abdullah University of Science and Technology (KAUST). Figures 1a, 2c, f, and 3a were created by Heno Hwang, a scientific illustrator at KAUST.

## Author contributions

S.H. conducted the study and drafted the manuscript. M.D.V. co-supervised the research and wrote the manuscript with inputs from all authors. M.D.V., A.T., and E.B. designed and conducted the crossflow filtration experiments for the membrane performance evaluation using realistic brines and performed the surface z-potential analysis. R.G. performed the Cryo-TEM and TEM experiments. B.D. conducted the X-ray crystallography and powder XRD experiments. X.L. assisted in the SEM measurements. S.N. conceived, supervised the work, and revised the manuscript. All authors contributed to the discussion of results and provided comments on the manuscript.

## Competing interests

The authors declare no competing interests.
