## [Peer Review File · Nature Communications]

Reviewers' Comments:

Reviewer #1:

Remarks to the Author:

This paper discloses a surprising and potentially very impactful extension of previous related work from the Nunes group toward high-performance separation membranes based on assembly of well-defined building blocks. The potentially high impact is based on the integration of self-assembled cyclodextrin aggregates into the selective barrier layer of nanofiltration membranes, what is significant under two aspects: i) it is performed using the industrially established interfacial polymerization toward thin polyamide layers on the porous support membrane, ii) it yields membranes that have a very high selectivity between lithium ions and magnesium or calcium ions, also at high feed concentrations and in different ratios what makes such membranes interesting for lithium recovery from brines. Overall, the work has been performed very systematically and using a wide range of methods that are state-of-the-art or cutting edge in the field, so that relevant fundamental insights are gained and that the engineering implication can also be assessed. The (main) paper is also very well written (the Supplementary Information will still need very careful editing in terms of language and formal correctness; apparently for this part emphasis until now was just on compiling the information).

A few points should be considered in the revision of the paper.

For me, the most critical questions are related to the real structure of the barrier layer and its formation (and control by fabrication parameters). I know that no full answer will be possible, but I miss things that could be done or discussed.

Different experiments and methods provide evidence that the partially aminated cyclodextrin derivatives with 7 glucose rings assemble into supramolecular arrays ("channel" like) and even form crystallites (or crystalline domains) under conditions that are then used for the interfacial polymerization. The work has somehow focused on those molecules (am7CD), and not so much on the partially aminated cyclodextrin derivatives with 6 or 8 glucose rings, because they lead to the best membranes for the purpose of the study (as indicated by the comparison in Fig. 2a). Is that because this particular structure (am7CD) forms more stable aggregates or just because less effort have been devoted to the am6CD and am8CD experiments?

In l. 170f. it is just mentioned that am6CD is not good, "likely due to the formation of defective layer".

This leads to the aspects I find puzzling and not yet fully clear. The evoked supramolecular channels exist in the aqueous phase, but the reaction between the amino-functional building blocks and TMC happens at the interface between aqueous and oil phase (see l. 104-5). For the polyamide formation to happen most efficiently, the orientation of the channels would be in plane of the layer. However, for the selective transport through the barrier layer, the channels should be orthogonal to the plane (but in such case only the upper rim of the channel would react ...). The membranes with the best performance have a barrier layer thickness of 15 nm, i.e. many building blocks should be aligned (and have a suited orientation in the layer) to enable the transport shown in the cartoon (Fig. 2c). On the other hand, I would also expect that the reaction of amino groups toward amide bonds in a network will interfere with the assembly (the cartoons in Figure 2c and 2f do not show a TMC-based "matrix"). Even when crystallites (with an order like shown in SI Figure 7) "survive" the reaction, there will also be less ordered and amorphous (polyamide) domains in the selective layer.

Hence, I am somewhat surprised that the characterization of the membranes has been exclusively performed by using salts (where the ionic character of the solute has major influences on the different mechanisms, including dehydration as shown in the paper). I would expect that in addition also a rigorous characterization of the nanofiltration membranes with neutral solutes will be performed, in order to either quantitatively demonstrate that the barrier pore size distribution is indeed only due to the self-assembled channels (and/or to elucidate the influence of other pathways including defects

through the barrier).

Smaller issues

l. 153: In the TEM image I do not see the evoked distance.

l. 241: It is good that the authors compare performance with literature and industrial state-of-the-art. In the discussion of Fig. 2h they just summarize that "membranes proposed in this study outperformed the other membranes". However, in the (very little) graph one can find one data point for a membrane with selectivity >300 and permeance of about 8 LMH/bar. I would rank that at least on par with what has been accomplished in the present work.

l. 254: I disagree that what is written is the typical sequence (of salt rejection) for a negatively charged membrane. Such membrane would certainly not have the highest rejection for MgCl₂! Sulfate salts would have the highest rejection.

Formal issues

l. 47: Incomplete sentence

l. 133: two times "ultrafiltration"

Reviewer #2:

Remarks to the Author:

The authors present a self-assembled cyclodextrin nano-channels in polyamide membrane for precise separation of Li⁺/Mg²⁺ ions. The intrinsic cavity of cyclodextrin is close to the size of Li⁺ and Mg²⁺ ions, the integration of formed nano-channels with accurate molecular dimension amplifies the dehydration effect and thus achieve high selectivity. The results are interesting and well organized. It can be published after some minor questions and corrections are sorted out. Some comments:

1) The cyclodextrin monomer used in this paper have been used in the authors' previous paper (Adv. Funct. Mater. 2020, 30, 1906797), what is the main difference between these two works?

2) How far apart are the two cyclodextrin channels? How does this distance affect the separation process?

3) To support the proposed concept, it would be helpful if the authors could compare and analyze the GI-WAXS results of the membranes with the self-assembled Am7CD powder and the simulated pattern from the single crystal data.

4) Since surface charge distribution is relevant for Li⁺/Mg²⁺ separation, it would be interesting if the authors could measure and compare the surface zeta potential of the LiOH-Am7CD and HAc-Am7CD membranes. If HAc-Am7CD membranes have more negative surface charge, one might expect that their selectivity would be lower.

5) The HAc-Am7CD membranes were fabricated with higher TMC concentration and reaction time, please explain this.

6) If I understand correctly, the optimized membrane was fabricated with a TMC concentration of 0.01%, while the membranes selected for practical brine tests were fabricated at 0.05%. Do the authors have some explanations on this?

7) Compared to the binary mixture, the simulated brine tests have much lower selectivity. Are there any potential reasons behind the authors could provide?

8) In Figure 1A, please increase the font size of the left legend and add a missing arrow; Please remove a duplicate in the right down legend; Please explain what the yellow and blue spheres in the Am7CD molecules represent.

9) In Figure 1F, please delete a duplicated scale bar.

10) Please explain what the dashed lines in Figure 3B mean.

REVIEWER COMMENTS

Reviewer #1 (Remarks to the Author):

1. This paper discloses a surprising and potentially very impactful extension of previous related work from the Nunes group toward high-performance separation membranes based on assembly of well-defined building blocks. The potentially high impact is based on the integration of self-assembled cyclodextrin aggregates into the selective barrier layer of nanofiltration membranes, what is significant under two aspects: i) it is performed using the industrially established interfacial polymerization toward thin polyamide layers on the porous support membrane, ii) it yields membranes that have a very high selectivity between lithium ions and magnesium or calcium ions, also at high feed concentrations and in different ratios what makes such membranes interesting for lithium recovery from brines. Overall, the work has been performed very systematically and using a wide range of methods that are state-of-the-art or cutting edge in the field, so that relevant fundamental insights are gained and that the engineering implication can also be assessed. The (main) paper is also very well written (the Supplementary Information will still need very careful editing in terms of language and formal correctness; apparently for this part emphasis until now was just on compiling the information).

Authors reply: We thank the reviewer for the positive and valuable comments. We have carefully revised the main text and the Supplementary Information accordingly. The changes in the manuscript are highlighted. Please find below a point-by-point response to the comments and concerns.

A few points should be considered in the revision of the paper.

2. For me, the most critical questions are related to the real structure of the barrier layer and its formation (and control by fabrication parameters). I know that no full answer will be possible, but I miss things that could be done or discussed.

Authors reply: We agree and acknowledge that this is important although there is no full answer to explain the kinetic formation of the barrier layer with the available characterization techniques. Through experimental evidence and results, we aimed to thoroughly investigate every aspect of the fabrication procedure to best delve into the formation of the membrane active layer, recognizing that not an absolutely complete picture can be provided.

3. Different experiments and methods provide evidence that the partially aminated cyclodextrin derivatives with 7 glucose rings assemble into supramolecular arrays (“channel” like) and even form crystallites (or crystalline domains) under conditions that are then used for the interfacial polymerization. The work has somehow focused on those molecules (am7CD), and not so much on the partially aminated cyclodextrin derivatives with 6 or 8 glucose rings, because they lead to the best membranes for the purpose of the study (as indicated by the comparison in Fig. 2a). Is

that because this particular structure (am7CD) forms more stable aggregates or just because less effort have been devoted to the am6CD and am8CD experiments?

Authors reply: We thank the reviewer for the question and we apologize for any confusion. Please note that the same effort was devoted to studying the three different aminated cyclodextrins.

Specifically, we polymerized Am6CD, Am7CD, and Am8CD molecules with increasing cavity sizes under the same fabrication conditions and we expected that the selectivity would increase as the cavity size decreased. However, as shown in Fig. 2a of the main text, the membranes fabricated with the smallest cavity (Am6CD) exhibited the lowest MgCl₂ and LiCl selectivity with a higher measured flux (Supplementary Figs. 17-19). We believe there are steric factors influencing the reaction leading at least in part to Am6CD defective membranes. Compared with Am7CD and Am8CD, Am6CD molecules have less reactive amino groups on the narrower rim, thus they have more spatial hindrance to react with the acyl chlorides, which tends to react with water instead. After performing several experiments at lower pH, we believe that the reaction between Am6CD and TMC is less efficient. In contrast, in the case of Am7CD and Am8CD molecules, they have higher reactivity to compete with water and react more efficiently with TMC to form a dense selective layer (Supplementary Fig. 10c, d). As a result, the rejection of MgCl₂ was consistently > 90% for both membranes (Supplementary Fig. 18).

As suggested in step#5 and to better assess the role of the cavity size, membrane porosity was further characterized using neutral solutes as markers in additional nanofiltration tests. In these experiments, a cut-off rejection greater than 90% reflects the effective pore size of membranes. For the sieving performance of Am6CD, Am7CD, and Am8CD membranes, please refer to the updated Supplementary Fig. 15 (answer to #step5). Here, the solutes with a hydrodynamic radius smaller than the cavity size permeated through the pores except for the Am6CD membranes, which were defective. While Am8CD and Am7CD membranes showed a threshold of 0.45 nm and 0.34 nm, respectively. Therefore, it is reasonable that for Am8CD membranes, the spatial resistance to the passage of Mg²⁺ and Li⁺ ions was accordingly lower than that observed for Am7CD membranes, leading to lower selectivity and a higher measured flux (Fig. 2a and Supplementary Fig. 19).

Ultimately, we would like to emphasize the importance of selectivity in ion separation applications. For Am8CD membranes, the separation factor was only 34, while it achieved 84 for Am7CD membranes. Therefore, this was one more reason to selected Am7CD molecules for more in-depth study.

Revisions made:

I. 168-I. 170: The Am6CD membranes exhibited the lowest separation factor and rejection of MgCl₂, probably due to the formation of a defective layer caused by the insufficient reactivity in competing with water to react with TMC (Supplementary Fig. 10).

I. 171-I. 174: Therefore, Am7CD-based membranes had the best performance which may be attributed to their optimal cavity size towards the target solutes and the sufficient reactivity with TMC under the membrane fabrication conditions.

4. In I. 170f. it is just mentioned that am6CD is not good, “likely due to the formation of defective layer”.

This leads to the aspects I find puzzling and not yet fully clear. The evoked supramolecular channels exist in the aqueous phase, but the reaction between the amino-functional building blocks and TMC happens at the interface between aqueous and oil phase (see I. 104-5). For the polyamide formation to happen most efficiently, the orientation of the channels would be in plane of the layer. However, for the selective transport through the barrier layer, the channels should be orthogonal to the plane (but in such case only the upper rim of the channel would react ...). The membranes with the best performance have a barrier layer thickness of 15 nm, i.e. many building blocks should be aligned (and have a suited orientation in the layer) to enable the transport shown in the cartoon (Fig. 2c). On the other hand, I would also expect that the reaction of amino groups toward amide bonds in a network will interfere with the assembly (the cartoons in Figure 2c and 2f do not show a TMC-based “matrix”). Even when crystallites (with an order like shown in SI Figure 7) “survive” the reaction, there will also be less ordered and amorphous (polyamide) domains in the selective layer.

Authors reply: We thank the reviewer for the comment.

Indeed, we could not experimentally clarify the precise orientation of the channels during the condensation reaction, but we would like to discuss our hypotheses about it as much as possible based on our findings.

The amino groups on the narrower rim of Am7CD molecules are more reactive than the hydroxyl groups on the wider rim¹. When the Am7CDs diffuse to the interface between the two immiscible phases, the narrow rim with highly reactive amino groups would preferentially face upwards towards the organic phase to enable the polycondensation reaction². Moreover, at the interface between the hydrophilic phase (water) and the hydrophobic phase (Isopar G), the 14 hydrophilic hydroxyl groups on the wider rim would advantageously orient face downward towards the aqueous phase. Therefore, we tend to believe that the channels can orient orthogonally.

We also agree with the reviewer that the amide formation efficiency are higher if the channels are in the plane of the interface. However, since TMC has little solubility in water, the polymerization occurs predominantly on the organic side of the interface, in which there are full of highly reactive TMC. Once the amine monomers are introduced into an excess of acyl chloride in the organic phase next to the interface, the amino groups could react effectively with the surrounding TMC molecules^{3,4}. In this case, we believe that the difference of reaction efficiency, caused by the orientation of the channels, could be neglected.

On the other hand, as confirmed by Cryo-TEM (Fig. 1b), there are ordered aggregates (crystallites) in the aqueous solution. These crystallites could diffuse at the interface where the condensation reaction occurs to be seamlessly embedded in the membrane active layer, as also confirmed by AFM and TEM analyses (Fig. 1e, f).

We also agree that the condensation reaction between TMC and the amino groups affects the channel structure unavoidably. However, we believe that the self-assembled channels will not be completely disrupted during the interfacial polymerization. It is, in fact, very difficult for the TMC to diffuse inside solid crystallites to enable the condensation reaction and most of the TMC only likely react with the outer Am7CD molecules of the crystallites. Therefore, the inner ordered structure would be preserved. Furthermore, X-ray crystallography data indicate that the head-to-tail channel structure are formed mainly driven by the intermolecular hydrogen bonding between electron-donors (amino groups on the narrower rim) and acceptors (hydroxyl groups on the wider rim). So, during the condensation, most of the amino groups will react with TMC, while some will remain unreacted due to the steric hindrance on the narrower rim. These unreacted amino groups could continue maintaining the existing hydrogen bonding frameworks. Furthermore, the amides formed from reacted amino groups potentially have the ability to form amide-hydroxyl hydrogen bonds with the neighboring hydroxyl groups^{5, 6}. In this case, the supramolecular channel structure should be mainly preserved even after the condensation reaction occurs, inevitably with some possible distortions or deformations.

Please note that in Fig.2c and 2f, we omitted the TMC to simplify the transport mechanism representation and convey the main idea to the readers.

5. Hence, I am somewhat surprised that the characterization of the membranes has been exclusively performed by using salts (where the ionic character of the solute has major influences on the different mechanisms, including dehydration as shown in the paper). I would expect that in addition also a rigorous characterization of the nanofiltration membranes with neutral solutes will be performed, in order to either quantitatively demonstrate that the barrier pore size distribution is indeed only due to the self-assembled channels (and/or to elucidate the influence of other pathways including defects through the barrier).

Authors reply: We thank the reviewer for the kind suggestion. Please refer to the updated Supplementary Fig. 15 for the characterization of the different types of nanofiltration membranes with neutral solutes. Specifically, a series of neutral sugars (including raffinose, sucrose, glucose, xylose, glycerol, and ethylene glycol) with different Stokes radii were dissolved in deionized water as markers to investigate the sieving performance of LiOH-**Am6CD**-0.1 TMC, LiOH-**Am7CD**-0.1 TMC and LiOH-**Am8CD**-0.1 TMC membranes. The radii calculations are also reported in the SI (the section titled 'Filtration tests of neutral solutes'). For these experiments, it can be assumed that a cut-off rejection greater than 90% reflects the effective pore size of the membrane⁷⁻⁹.

Please find below also the results for the LiOH-Am7CD-0.05 TMC tested under the same experimental conditions (Supplementary Fig. 16). Here, it is possible to observe a cut-off at about 0.33 nm, which is consistent with the previous findings.

Revisions made:

Supplementary Figure 15. Rejection profile of neutral solutes (including raffinose, sucrose, glucose, xylose, glycerol, and ethylene glycol) plotted against the calculated solute Stokes radius for the three types of cyclodextrin membranes, respectively. The grey horizontal dashed line denotes a rejection cut-off of 90%.

Supplementary Figure 16. Observed rejection of neutral solutes (including raffinose, sucrose, glucose, xylose, glycerol and ethylene glycol) plotted as a function of the calculated solute Stokes radius for the LiOH-Am7CD-0.05 TMC membranes. The grey horizontal dash line indicates a rejection cut-off of 90%.

Smaller issues

I. 153: In the TEM image I do not see the evoked distance.

Authors reply: Please refer to the revised Fig. 1f in which the new chosen zoom factor (scale bar 1.5 nm) makes the evoked distance easily distinguishable.

Revisions made:

f.

Fig. 1f TEM images of Am7CD membranes with incorporated assemblies (LiOH-Am7CD-0.05 TMC), highlighting a high magnification detail. Inset (with a scale-bar of 1.5 nm): the magnification of a selected area showing distinct lattice fringes.

I. 149-I. 152: AFM analysis corroborates these results and shows that the crystals were uniformly distributed throughout the membrane (Fig. 1e), while TEM images of the polyamide selective layer reveal long-range ordering of these crystals with a distance of 2.53 Å between the adjacent lattice fringes (inset of Fig. 1f, scale bar 1.5 nm).

I. 241: It is good that the authors compare performance with literature and industrial state-of-the-art. In the discussion of Fig. 2h they just summarize that “membranes proposed in this study outperformed the other membranes”. However, in the (very little) graph one can find one data point for a membrane with selectivity >300 and permeance of about 8 LMH/bar. I would rank that at least on par with what has been accomplished in the present work.

Authors reply: We thank the reviewer for the kind suggestion. We have revised the text accordingly and modified the graph as we understood it was not clear that the red stars were relative to 3 different membranes and not clear to which type of membranes we were referring to.

Revision made:

I. 242–I. 244: Remarkably, LiOH-Am7CD-0.01 TMC membranes proposed in this study outperformed most of the previously reported membranes, exhibiting superior selectivity (>800) and moderate permeance.

I. 254: I disagree that what is written is the typical sequence (of salt rejection) for a negatively charged membrane. Such membrane would certainly not have the highest rejection for MgCl₂! Sulfate salts would have the highest rejection.

Authors reply: We thank the reviewer for pointing out this mistake. We have corrected this editing error in the revised manuscript, which is now in agreement with our experimental results (Supplementary Fig. 29).

Revision made:

I. 257-I. 259: Therefore, the rejection rates followed the typical sequence of negatively charged nanofiltration membranes, which is Na₂SO₄ > MgSO₄ > MgCl₂ > NaCl (Supplementary Fig. 29).

Formal issues

I. 47: Incomplete sentence

I. 133: two times “ultrafiltration”

Authors reply: We thank the reviewer for carefully reading the manuscript. We have corrected the errors in the revised manuscript.

Revisions made:

I. 46-I. 47: Effective and sustainable methods for the recovery of metal ions from different sources including e-waste recycling are increasingly important.

I. 129-I. 131: Scanning electron microscopy (SEM) (Supplementary Figs. 9-11) and atomic force microscopy (AFM) (Supplementary Figs. 12 and 13) revealed a smooth and continuous defect-free active layer on the ultrafiltration polyacrylonitrile (PAN) support layer.

Reviewer #2 (Remarks to the Author):

The authors present a self-assembled cyclodextrin nano-channels in polyamide membrane for precise separation of Li⁺/Mg²⁺ ions. The intrinsic cavity of cyclodextrin is close to the size of Li⁺ and Mg²⁺ ions, the integration of formed nano-channels with accurate molecular dimension amplifies the dehydration effect and thus achieve high selectivity. The results are interesting and well organized. It can be published after some minor questions and corrections are sorted out. Some comments:

Authors reply: We thank the reviewer for the valuable comments. A point-by-point response to the comments and concerns is reported below. The changes within the manuscript are highlighted.

1) The cyclodextrin monomer used in this paper have been used in the authors' previous paper (Adv. Funct. Mater.2020, 30, 1906797), what is the main difference between these two works?

Authors reply: We thank the reviewer for the question.

The first main difference is that in our work, functionalized cyclodextrins were self-assembled into channels to participate in the polycondensation reaction and be incorporated into the selective layer to provide precise ion separation. In this case, the cavities of the cyclodextrins were oriented via supramolecular interactions, whereas the previously reported work focused more on using the intrinsic cavity of the cyclodextrins to achieve shape selectivity in organic solvent nanofiltration and the cavities were more randomly cross-linked in the plane during synthesis.

Another main difference is the use of a more reactive cross-linker, namely, trimesoyl chloride (TMC). In particular, to achieve target ion separation performance, the degree of cross-linking is critical. Consequently, the higher reactivity of TMC leads to the formation of a denser layer with excellent ion separation properties. In contrast, the other work used a less reactive cross-linker (terephthaloyl chloride) to produce an ultrathin layer, whose properties are accordingly different.

Finally, the new manuscript is fully focused on ions separations and testing with simulated brines, while the previous paper aimed organic solvent nanofiltration.

2) How far apart are the two cyclodextrin channels? How does this distance affect the separation process?

Authors reply: We thank the reviewer for the valuable comment. We used *Olex2* software to calculate the volume of the channel cavities and the voids between the channels. The results showed that the voids have a volume of about 370 \AA^3 , which was similar to the cavity volume (360 \AA^3). Therefore, we believe that these voids can act like channel cavities, selectively filtering Mg^{2+} ions and allowing Li^+ ions to pass through based on the spatial resistance.

3) To support the proposed concept, it would be helpful if the authors could compare and analyze the GI-WAXS results of the membranes with the self-assembled Am7CD powder and the simulated pattern from the single crystal data.

Authors reply: We appreciate the suggestion from the reviewer. The figure has been updated following the reviewer's comment. As shown in the updated Supplementary Fig. 14, the simulated diffraction pattern of the stacked Am7CD channel well matches with the X-Ray diffraction data of bulk Am7CD polycrystalline powder as well as the integrated 1D in- and out-of-plane XRD patterns from GIWAXS data. The GIWAXS result shows the presence of polycrystalline species (lines in GIWAXS) and small crystallites (dots in GIWAXS). All the reflections are clearly matching the Am7CD crystal. Therefore, it is evident that the crystalline nature as well as self-assembled channels within the crystallites of the Am7CD were preserved in the membrane after the condensation reaction. However, it is worth mentioning that the small crystallites are less ordered with respect to each other^{10,11}. The text in the manuscript was slightly modified to reflect the reviewer's comment.

Revisions made:

I. 146-I. 147: Notably, the GIWAXS analysis indicates the presence of polycrystalline species and small crystallites over the membrane selective layer.

I. 154-I. 158: Furthermore, the good agreement between the GIWAXS in-plane and out-of-plane scattering profiles and the powder X-ray diffraction of the self-assembled Am7CD polycrystalline powder, in conjunction with the calculated XRD pattern, indicates the successful integration of the subnanometer channel structure within the membrane (Supplementary Fig. 14).

Supplementary Figure 14. Comparative spectral analysis including extracted GIWAXS one-dimensional spectra profiles (black and red curves) of LiOH-Am7CD-0.01 TMC membrane, PXRD results of polycrystalline Am7CD powder (blue curve) and the simulated pattern of stacked Am7CD (green curve).

4) Since surface charge distribution is relevant for Li⁺/Mg²⁺ separation, it would be interesting if the authors could measure and compare the surface zeta potential of the LiOH-Am7CD and HAC-Am7CD membranes. If HAC-Am7CD membranes have more negative surface charge, one might expect that their selectivity would be lower.

Authors reply: We thank the reviewer for the kind comments and suggestions. We have performed additional surface zeta-potential tests to compare the trend obtained for HAC-Am7CD membranes with that of LiOH-Am7CD membranes. As a result, HAC-Am7CD membrane surface is less negatively charged than LiOH-Am7CD membrane and this is possibly because less TMC hydrolyzed to carboxylic acid with negative charges at a lower pH during synthesis. Therefore, since the selectivity performance of both membranes was measured under the same filtration conditions (feed solution, temperature, operation pressure, etc.), the observed lower selectivity of HAC-Am7CD membranes is not a result of the Donnan effect.

Supplementary Fig. 26 has been added to the SI and the experimental result is also mentioned in the main text as follows:

Manuscript:

I. 225-I. 227: Note that membranes prepared with HAc and LiOH had similar crosslinking degree, surface morphology, thickness and surface charge (Supplementary Figs. 7-13, 25, 26).

Supplementary Information:

Supplementary Figure 26. Surface z-potential of HAc-Am7CD-0.3 TMC membrane plotted against solution pH at 1mM electrolyte concentration.

5) The HAc-Am7CD membranes were fabricated with higher TMC concentration and reaction time, please explain this.

Authors reply: We thank the reviewer for the question. As widely reported in the literature, during interfacial polymerization, parameters such as the pH of the aqueous solution, the concentration of the acyl chloride, and the reaction time can influence the cross-linking degree and the thickness of the selective layer. Specifically, in our experiments, the pH value of the aqueous solution during fabrication was 7.4 and 8.4 for HAc-Am7CD and LiOH-Am7CD, respectively. Therefore, to ensure a fair comparison, we optimized the reaction time and TMC concentration to synthesize HAc-Am7CD membranes with thickness and degree of cross-linking comparable to those obtained for LiOH-Am7CD membranes.

6) If I understand correctly, the optimized membrane was fabricated with a TMC concentration of 0.01%, while the membranes selected for practical brine tests were fabricated at 0.05%. Do the authors have some explanations on this?

Authors reply: We thank the reviewer for the observation and question. The choice of membranes for testing practical brines is mainly related to the robustness of the active layer. In

fact, although the thinner membrane (fabricated with a TMC concentration of 0.01%w/v) reported the highest selectivity at lower applied pressure and salinity, the thickness of the active layer was only 15 nm and might not ensure the necessary mechanical strength during the filtration of the synthetic brine solutions, in which extended operational times, large cross-flow velocity values, and high applied pressure (always >60 bar) were required due to their elevated salinity. Thus, we selected the membranes fabricated with a TMC concentration of 0.05%w/v, which had a thickness of 29 nm instead, resulting in a more mechanically robust layer.

7) Compared to the binary mixture, the simulated brine tests have much lower selectivity. Are there any potential reasons behind the authors could provide?

Authors reply: Compared with the binary feed solutions containing only 500 ppm of Mg^{2+} and 15 ppm Li^+ , the simulated brine and the other three synthetic feed solutions, mimicking real lithium-containing brines (i.e., concentrated seawater, Lungmu Co salt-lake brine, and Imperial geothermal brine) exhibited a considerably higher complexity in terms of both composition and ionic ratios (see Supplementary Table 5). For example, the concentration of competitive monovalent ions, such as Na^+ , is substantially higher than that of Li^+ . Therefore, as our membranes cannot selectively separate monovalent ions, Na^+ ions will tend to occupy the channels of the active layer, hence reducing Li^+ transport and negatively affecting selectivity. In addition, the overall salinity in all three synthetic brines was 20-30 times higher than that used in the binary feed solutions. Therefore, the high concentration polarization as well as the ionic properties, such as hydrated radius and hydration free energy, affected the performance and reduce the selectivity^{12, 13}.

8) In Figure 1A, please increase the font size of the left legend and add a missing arrow; Please remove a duplicate in the right down legend; Please explain what the yellow and blue spheres in the Am7CD molecules represent.

Authors reply: We thank the reviewer for the suggestion, Figure 1a has been modified accordingly.

9) In Figure 1F, please delete a duplicated scale bar.

Authors reply: We apologize with the reviewer for reporting a duplicated scale bar. Figure 1f has been corrected.

f.

10) Please explain what the dashed lines in Figure 3B mean.

Authors reply: We thank the reviewer for the suggestion. We have added a description in the caption of Figure 3b to explain that the dash line is simply a guide for the reader.

Revision made:

Fig. 3 Separation mechanism. **a**, Scheme illustrating different ions passing through or rejected by ordered LiOH-Am7CD-0.05 TMC membranes. **b**, **c**, **d**, Observed rejection versus **(b)** ionic diameter, **(c)** hydrated diameter, and **d**, molar Gibbs energy of hydration of different cations (with Cl⁻ as the counter-ion in the solution). Filtration tests were performed using a series of single-salt feed solutions with a concentration of 2000 ppm. The dash line is simply a guide for the reader.

- (1) Huang, T.; Puspasari, T.; Nunes, S. P.; Peinemann, K. V. Ultrathin 2D-layered cyclodextrin membranes for high-performance organic solvent nanofiltration. *Advanced Functional Materials* **2020**, *30* (4), 1906797.
- (2) Jiang, Z.; Dong, R.; Evans, A. M.; Biere, N.; Ebrahim, M. A.; Li, S.; Anselmetti, D.; Dichtel, W. R.; Livingston, A. G. Aligned macrocycle pores in ultrathin films for accurate molecular sieving. *Nature* **2022**, *609* (7925), 58-64.
- (3) Tan, Z.; Chen, S.; Peng, X.; Zhang, L.; Gao, C. Polyamide membranes with nanoscale Turing structures for water purification. *Science* **2018**, *360* (6388), 518-521.
- (4) Morgan, P. W.; Kwolek, S. L. Interfacial polycondensation. II. Fundamentals of polymer formation at liquid interfaces. *Journal of Polymer Science* **1959**, *40* (137), 299-327.
- (5) Harings, J. A.; Yao, Y.; Graf, R.; van Asselen, O.; Broos, R.; Rastogi, S. Erasing conformational limitations in N, N'-1, 4-Butanediyl-bis (6-hydroxy-hexanamide) crystallization from the superheated state of water. *Langmuir* **2009**, *25* (13), 7652-7666.
- (6) Habermann, S. M.; Murphy, K. P. Energetics of hydrogen bonding in proteins: A model compound study. *Protein Science* **1996**, *5* (7), 1229-1239.
- (7) Wang, Z.; Wang, Z.; Lin, S.; Jin, H.; Gao, S.; Zhu, Y.; Jin, J. Nanoparticle-templated nanofiltration membranes for ultrahigh performance desalination. *Nature communications* **2018**, *9* (1), 2004.
- (8) Liang, Y.; Zhu, Y.; Liu, C.; Lee, K.-R.; Hung, W.-S.; Wang, Z.; Li, Y.; Elimelech, M.; Jin, J.; Lin, S. Polyamide nanofiltration membrane with highly uniform sub-nanometre pores for sub-1 Å precision separation. *Nature communications* **2020**, *11* (1), 2015.
- (9) Zhao, C.; Zhang, Y.; Jia, Y.; Li, B.; Tang, W.; Shang, C.; Mo, R.; Li, P.; Liu, S.; Zhang, S. Polyamide membranes with nanoscale ordered structures for fast permeation and highly selective ion-ion separation. *Nature Communications* **2023**, *14* (1), 1112.
- (10) Topchieva, I. N.; Tonelli, A. E.; Panova, I. G.; Matuchina, E. V.; Kalashnikov, F. A.; Gerasimov, V. I.; Rusa, C. C.; Rusa, M.; Hunt, M. A. Two-phase channel structures based on α -cyclodextrin- polyethylene glycol inclusion complexes. *Langmuir* **2004**, *20* (21), 9036-9043.
- (11) Takahashi, S.; Yamada, N. L.; Ito, K.; Yokoyama, H. Inclusion Complex of α -Cyclodextrin with Poly (ethylene glycol) Brush. *Macromolecules* **2016**, *49* (18), 6947-6952.
- (12) Tansel, B. Significance of thermodynamic and physical characteristics on permeation of ions during membrane separation: Hydrated radius, hydration free energy and viscous effects. *Separation and purification technology* **2012**, *86*, 119-126.
- (13) Eigen, M.; Wicke, E. The thermodynamics of electrolytes at higher concentration. *The Journal of Physical Chemistry* **1954**, *58* (9), 702-714.

Reviewers' Comments:

Reviewer #1:

Remarks to the Author:

The authors have very well addressed the comments and questions, also by performing additional experiments. The paper could now be accepted.

Reviewer #2:

Remarks to the Author:

Authors answered all questions from reviewers very well, and revised the manuscript based on comments. I think the updated manuscript can be accepted now.